# Water nanolayer facilitated solitary-wave-like blisters in MoS₂ thin films

Enze Wang[1,4], Zixin Xiong[2,4], Zekun Chen[2], Zeqin Xin[1], Huachun Ma ⬡[2], Hongtao Ren[3], Bolun Wang[1], Jing Guo[1], Yufei Sun[1], Xuewen Wang[1], Chenyu Li[1], Xiaoyan Li ⬡[2] ✉ & Kai Liu ⬡[1] ✉

Solitary waves are unique in nonlinear systems, but their formation and propagation in the nonlinear fluid-structure interactions have yet to be further explored. As a typical nonlinear system, the buckling of solid thin films is fundamentally related to the film-substrate interface that is further vulnerable to environments, especially when fluids exist. In this work, we report an anomalous, solitary-wave-like blister (SWLB) mode of MoS₂ thin films in a humid environment. Unlike the most common telephone-cord and web buckling deformation, the SWLB propagates forward like solitary waves that usually appear in fluids and exhibits three-dimensional expansions of the profiles during propagation. In situ mechanical, optical, and topology measurements verify the existence of an interfacial water nanolayer, which facilitates a delamination of films at the front side of the SWLB and a readhesion at the tail side owing to the water nanolayer-induced fluid-structure interaction. Furthermore, the expansion morphologies and process of the SWLB are predicted by our theoretical model based on the energy change of buckle propagation. Our work not only demonstrates the emerging SWLB mode in a solid material but also sheds light on the significance of interfacial water nanolayers to structural deformation and functional applications of thin films.

Solitary waves are unique ordered wave structures formed in nonlinear systems and fundamentally arise from a delicate balance between dispersion and nonlinear effects[1–3]. Since their discovery in fluids, it was an important and interesting question in history whether solitary waves only appeared in fluids. During the past decades, there have been growing attention and some discoveries of solitary waves or solitons in solid-state physics[4,5], photonics[6,7], and neurodynamics[8], which suggests the existence of solitary waves or solitons in non-fluid systems. When fluid interacts with deformable solids, whether the nonlinear effects in the fluid-structure interaction (FSI) could trigger solitary waves or solitons has been investigated by a combination of theoretical modelling and numerical calculations. Previous studies[9–11]

demonstrated that the numerical solutions of solitary wave can be obtained in the fluid-filled elastic tubes by neglecting the high-order terms. It indicates the existence of solitary waves in the FSI due to the combined effects of nonlinearity and weak dispersion. However, there has been no experimental study to report the formation and propagation of solitary waves in the FSI.

As a typical nonlinear system, the buckling of solid thin films usually follows meandering propagation modes, including straight-sided[12–14], circular[15,16], telephone-cord (TC)[17–19], ring-shaped[20,21], and web buckles[22,23], due to the mixed-mode interface fracture and pinning effects. The film-substrate interface is crucial for determining the buckling modes of the thin film, but it is vulnerable to diverse

[1]State Key Laboratory of New Ceramics and Fine Processing, School of Materials Science and Engineering, Tsinghua University, Beijing 100084, China. [2]Centre for Advanced Mechanics and Materials, Applied Mechanics Laboratory, Department of Engineering Mechanics, Tsinghua University, Beijing 100084, China. [3]School of Materials Science and Engineering, Liaocheng University, Liaocheng 252000, China. [4]These authors contributed equally: Enze Wang, Zixin Xiong. ✉e-mail: xiaoyanlithu@tsinghua.edu.cn; liuk@tsinghua.edu.cn

environmental changes. In particular, the interface often interacts with fluids that originate from humidity, organic solvents, pump oil, etc. The FSI triggers the buckling process at the interface and influences the dynamic propagations of buckles in a thin film. For example, TC buckles initially form in a titanium film deposited on a polymer with the addition of alcohol drops and later merge into larger branched straight-sided buckles[24]. Similarly, ring-shaped buckles appear in the weak adhesion zones that result from the coffee-ring effect of oil evaporation[21]. Moreover, fluid flowing through the buckled channels due to the capillary effect facilitates the synergistic interaction between the fluid and buckles, resulting in a much faster and further propagation of buckles[25]. Despite these discoveries, investigation of FSI-induced novel buckling modes is always intriguing and significant, as any buckling mode may have a fatal impact on the applications of thin films.

In this study, we report an anomalous, solitary-wave-like blister (SWLB) mode of $MoS_2$ thin films induced by interfacial FSI on a rigid substrate. Unlike the most common TC buckles and web buckles, arc buckles surprisingly emerge in $MoS_2$ films under high humidity and propagate forward like solitary waves that usually appear in fluids. These solitary-wave-like blisters exhibit three-dimensional expansions of the profiles during their propagation owing to the gradual release of residual elastic strain energy in the $MoS_2$ film, which have not been observed or reported previously. In situ mechanical, optical, and atomic force microscopy (AFM) measurements verify at high relative

humidity the existence of a 3-nm-thick interfacial water nanolayer, which modulates the film-substrate interface interaction and thus determines the SWLB propagation. A front delamination and tail re-adhesion mechanism is proposed to explain the dynamic propagation of the emerging SWLB mode, in which the profile-expanded propagation and the critical stop-point of the SWLB can be reasonably predicted by energy-based theoretical modeling. Our work not only demonstrates the emerging SWLB mode in a solid material but also sheds light on the significance of interfacial water nanolayers to the structural deformation and functional applications of thin films.

## Results

### Solitary-wave-like propagation of buckles

Solitary waves can localize energy, maintain their shapes, and propagate for a long distance[1] (Fig. 1a), as a result of the balanced dispersion by nonlinearity[2,3]. In contrast, ordinary waves are gradually attenuated during their propagation due to damping effects (Fig. 1b). In the systems of solid thin films, the SWLB mode refers to the deformed buckles propagating integrally instead of being pinned at the original region (Fig. 1c). The most obvious SWLB feature is the re-adhesion of the thin film onto the substrate at the tail side during propagation (Fig. 1d), which is distinct from ordinary buckling modes (Fig. 1e). In the latter, the delaminated region of the buckles remains detached from the substrate, while the front tips of the buckles continue to propagate forward, extending the buckling area (Fig. 1f).

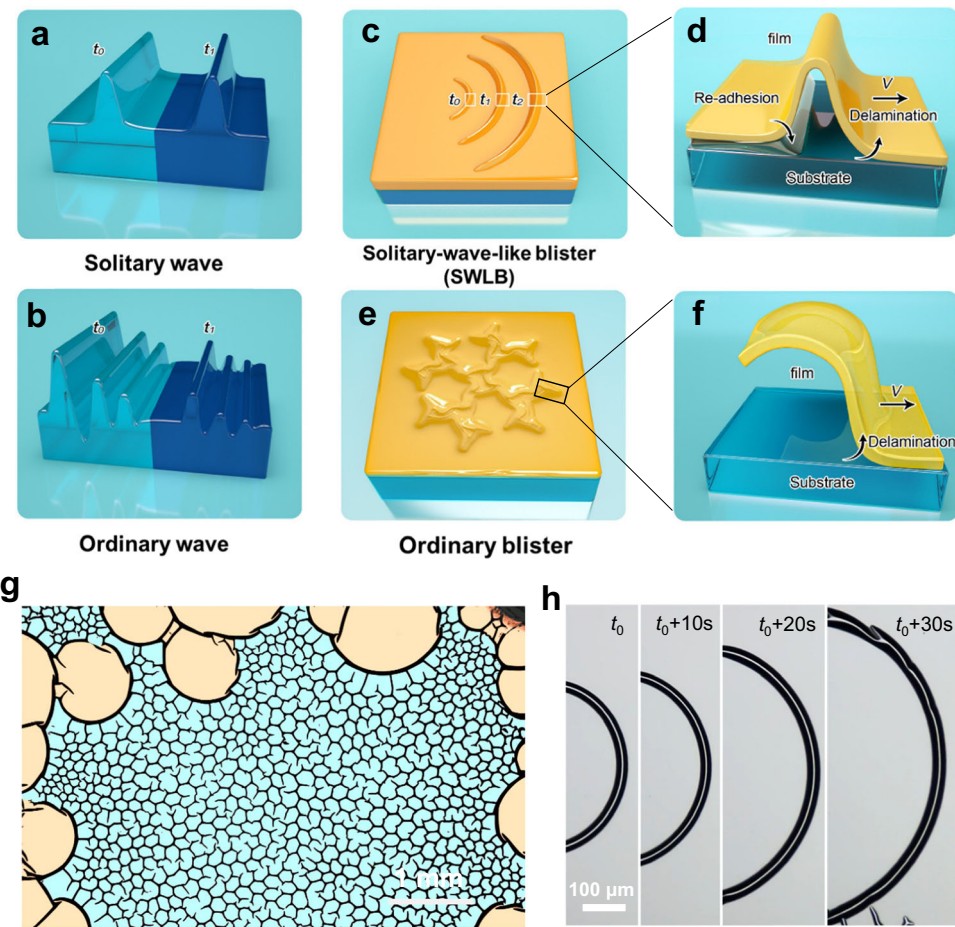

**Fig. 1 | Schematic illustrations and morphologies of the SWLB mode compared with the ordinary mode. a, b** Schematic illustrations of a solitary wave (**a**) and ordinary wave (**b**). **c, d** Schematic illustrations of the SWLB mode (**c**) and its cross-section in the propagation direction (**d**). **e, f** Schematic illustrations of an ordinary buckle (**e**) and its cross-section in the propagation direction (**f**). **g** False-color optical image of the formed SWLB (arc buckles, the orange region) and ordinary buckling (web buckles, the cyan region). The morphologies of SWLB and ordinary buckling are false-colored orange and blue, respectively. **h** Morphologies of a propagating SWLB.

The $MoS_2$ films used in this work were prepared by polymer-assisted deposition (PAD) similar to our previous work[23] (see details in Methods and Supplementary Fig. 1). The Raman spectrum collected from the strained film shows that the in-plane vibrational mode $E_{2g}^1$ of $MoS_2$ blueshifts by ~1.8 cm$^{-1}$ compared to that collected from the released $MoS_2$ film, where the stress should be fully relaxed. And the Raman spectrum collected from the film after SWLB propagated well match those of the same area peeled off, which indicates that the strain in the $MoS_2$ film is completely released after the SWLB propagated (Supplementary Fig. 1f). This blueshift of $E_{2g}^1$ suggests that the as-prepared $MoS_2$ film bears a biaxial compressive strain of ~0.35% in the $MoS_2$ film[26], which may result from the mismatch of thermal expansion between $MoS_2$ and sapphire during growth.

Under such a compressive strain, the as-prepared films remain flat in a dry atmosphere. However, when exposed to high relative humidity (RH, > 80%) in our homemade equipment (Supplementary Fig. 2), the region of a $MoS_2$ film around the edge of the substrate deforms into arc buckles propagating forward integrally (Supplementary Movie 1 and Fig. 1g), revealing the feature of the SWLB mode. After propagating for a certain distance, the arc buckles may fracture along the central line (donated as fractured buckles, Supplementary Fig. 3a) or break into several segments (donated as torn buckles, Supplementary Fig. 3b). Then, other arc buckles may nucleate at the cracks or the nodes of different segments, as shown in Supplementary Movie 1. Finally, common web buckles begin to root from the arc buckles and propagate towards the centre of the sample (Supplementary Fig. 3b and Supplementary Movie 1). To better demonstrate the SWLB feature, the morphology of one propagating SWLB was observed in situ (Supplementary Movie 2 and Fig. 1h). The radius of the SWLB increases gradually during propagation, while the area of the flat zone swept by the SWLB also expands. The propagation speed of arc buckles is several micrometers per second (Supplementary Fig. 4).

## Dimensional change of the SWLB during propagation

Profile changes of arc buckles during their propagation are indicators to reveal the buckling dynamics. By decreasing the humidity rapidly by shutting off the humidified gas and purging the sample with dry gas, the propagation of arc buckles can be frozen temporarily, and the buckles will continue to propagate when exposed to the high humidity again. This phenomenon provides a way to measure the profile changes of arc buckles at different stages of their propagation. As shown in Fig. 2a–f, one arc buckle propagates with a time-related, enlarging curvature radius ($R_t$) and finally deforms into a series of broken buckles, from which some web-like buckles begin to emerge. With the increase in $R_t$, both the transient height $\delta_t$ and the half-width $b_t$ of the buckles increase (Fig. 2g), which means the increases in the total interfacial area and in the elastic energy of the buckles. The relationship between $\delta_t$ and $R_t$, as well as between $b_t$ and $R_t$ is quite linear (Fig. 2h). Moreover, the final height ($\delta_f$) and final half width ($b_f$) of the arc buckles also obey a linear relationship (Supplementary Fig. 5). A similar linear relationship between $\delta$ and $b$ is also reported in straight-sided and ring-shaped buckles[21,27]. The expansion of the three-dimensional profile ($R_t$, $\delta_t$, $b_t$) and the total interfacial area of arc buckles suggests that the residual elastic strain energy stored in the

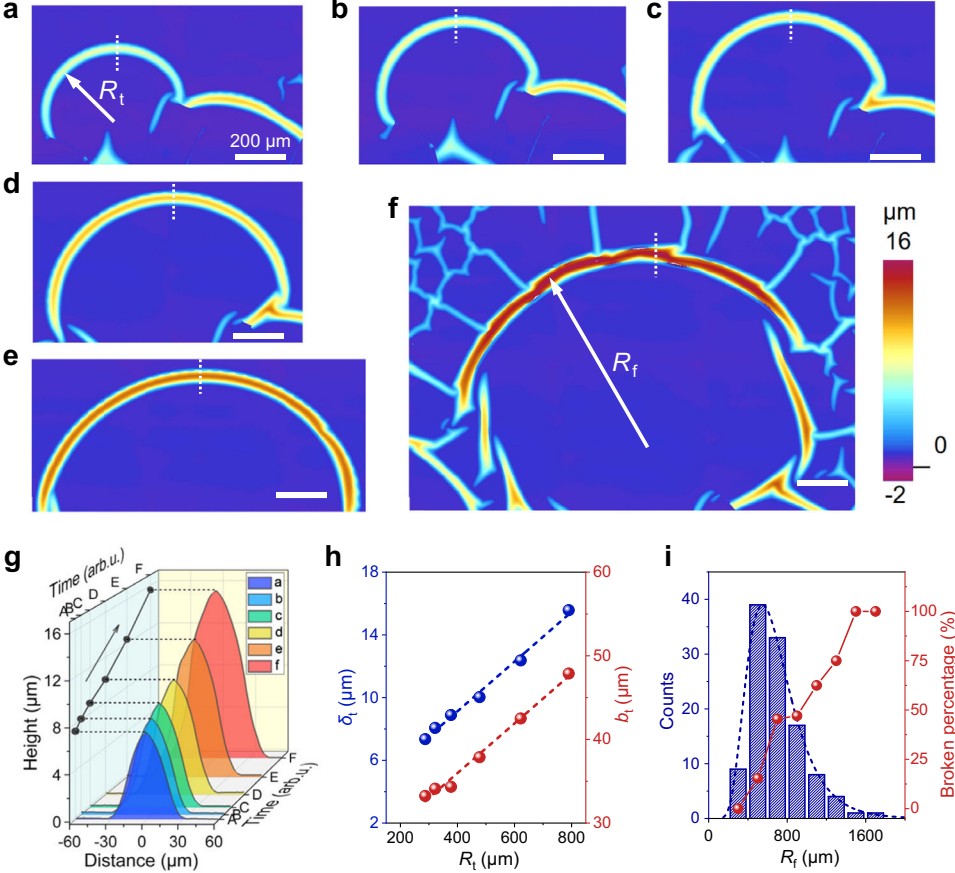

**Fig. 2 | Characteristic dimensions of one arc buckle at different moments during its propagation. a–f** Surface profiles of the arc buckles at different moments. $R_t$ is the transient radius of the arc buckle at a specific time $t$, and $R_f$ indicates the final radius of the arc buckle when the arc buckle no longer propagates. These two parameters can be measured experimentally. Scale bars, 200 μm. **g** Height-profile lines crossing the middle of arc buckles marked by white lines in figures **a–f**. **h** Dependence of the height $\delta_t$ and the half-width $b_t$ on the transient curvature radius $R_t$. **i** Statistical distribution of the final curvature radius $R_f$ and the broken percentage of arc buckles as a function of $R_f$. Source data are provided as a Source Data file.

MoS$_2$ film swept by the buckles may have been transferred to the buckles, leading to the gradual increase in the elastic energy of the buckles during propagation. To check whether the blister height of SWLB is beyond the post-buckling amplitude, we choose an arc buckle where web buckles form aside (Supplementary Fig. 6a), and measure the heights of the arc and web buckles. As shown in Supplementary Fig. 6b, the height of the arc buckle (SWLB) was ~14 μm, much larger than the heights of the web buckles (post-buckling blisters) ranging from 2 to 3.5 μm. This reveals that the height of arc blister is far beyond the post-buckling amplitude, showing the release and accumulation of more residual elastic strain energy.

Due to the redistribution of the film energy during SWLB propagation, the structural stability of the arc buckles is also related to their curvature radius (Fig. 2i). The final curvature radius ($R_f$) of the buckles mostly ranges from 400 to 1000 μm and approximately obeys the lognormal distribution. Since $R_f$ is also the propagation distance of an arc buckle, this index reflects the propagation ability of the SWLB. It is noted that with increasing $R_f$, there exists a higher percentage of broken buckles (including fractured buckles and torn buckles), especially for $R_f > 1000$ μm, suggesting the instability of large buckles. This radius-related instability will be explained subsequently by our theoretical model.

### Effects of humidity on the SWLB mode

It is well-accepted that the interface state is of vital importance for the formation and propagation of buckling deformations. Conventionally,

TC buckles and web buckles are the most common deformations for thin films on rigid substrates. The formation of abnormal SWLB deformation indicates a dramatic change in the film-substrate interface state under high humidity. To investigate the effect of humidity on the interface, the adhesion force between the MoS$_2$ film and the sapphire substrate was first measured (Supplementary Fig. 7). It is found that the adhesion strength decreases dramatically from 0.78 to 0.24 MPa when the RH increases from 60% to 80% at room temperature (Fig. 3a and Supplementary Table 1). Such a decrease in adhesion is also reported in SiO$_2$-SiO$_2$ nanoasperity contact with the water meniscus in a similar humidity range[28]. Meanwhile, the elastic modulus and hardness of the MoS$_2$ film remain almost the same after the film is exposed to high humidity for three days (Supplementary Fig. 8), which excludes the softening of the MoS$_2$ film by absorption or reaction with water molecules.

The reduced adhesion strength under high humidity indicates a weakened film-substrate interfacial interaction, which is ascribed to the hydrophilic nature of sapphire and MoS$_2$. Based on the Owen-Wendt-Rabel-Kaelble (OWRK) method[29,30], the contact angles and the adhesion work between the MoS$_2$ film and water and between sapphire and water are 86°, 0.077 J m$^{-2}$ and 16°, 0.142 J m$^{-2}$, respectively (Supplementary Section 9). The more positive adhesion work of sapphire means that water is more energetically favorable to spread on the sapphire surface. As a result, a water nanolayer will form on the sapphire surface due to capillary condensation under high humidity[31–35]. Similarly, a water layer may also form at the MoS$_2$-sapphire interface

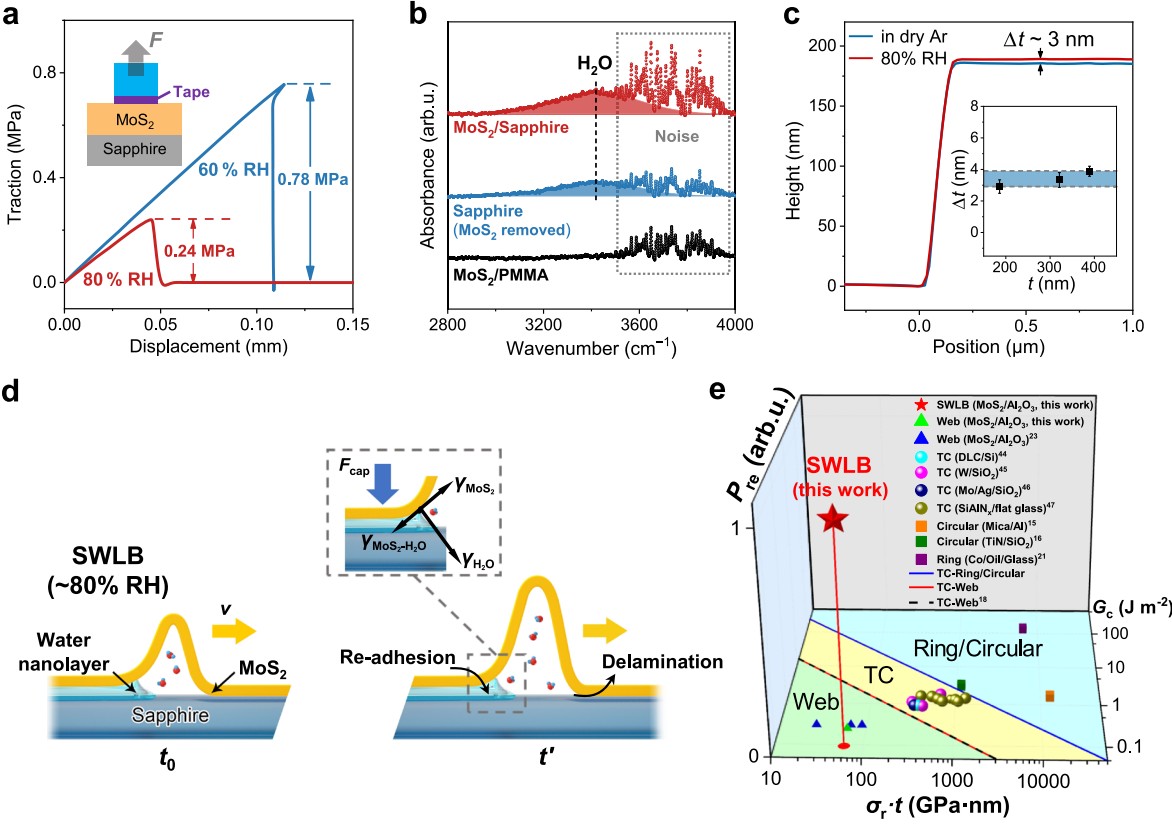

**Fig. 3 | Effects of humidity on the formation of the SWLB mode. a** Traction-displacement curve measured by in situ mechanical measurements under different humidities. The inset shows a schematic of the measurement apparatus. **b** Absorbance spectra measured by ATR-FTIR under high humidity. The fluctuations of the absorbance signals in the wavenumber range labelled by dotted lines originate from the noise of equipment. **c** Height profiles of the MoS$_2$ film in dry Ar and under 80% RH, measured by AFM. The inset shows that the mean height changes ($\Delta t$) range from 2.9 to 3.9 nm for different thicknesses ($t$) of films, where the data errors are standard deviations of 5 independent measurements for each thickness. **d** Mechanism of the humidity-driven SWLB mode. The inset shows a schematic illustration of the re-adhesion effect induced by the capillary force of the interfacial water nanolayer at the tail side of the SWLB. **e** Comparison between the SWLB mode and other ordinary buckling modes. Light cyan, yellow, and green regions represent ring/circular buckles, TC buckles, and web buckles, respectively. $G_c$ is the interface energy, and $\sigma_r \cdot t$ represents the membrane force that is the product of residual stress and film thickness. $P_{re}$ indicates whether there is (value = 1) or there is no (value = 0) readhesion process. Source data are provided as a Source Data file.

under high humidity. As shown in the in situ attenuated total reflection Fourier transform infrared (ATR-FTIR) spectroscopy (Fig. 3b), at ~80% RH, the $MoS_2$ film on sapphire shows a strong peak at ~3430 $cm^{-1}$ that corresponds to a disordered H-bonded structure, which is a feature of "liquid-like" water[36,37]. Since the penetration depth of ATR-FTIR is ~1 μm, the collected signals of water would include those adsorbed on the top surface of the $MoS_2$ film and at the $MoS_2$-sapphire interface. However, the absence of water peaks for the $MoS_2$ film on polymethyl methacrylate (PMMA) at the same humidity excludes water adsorption on the top surface of the $MoS_2$ film. Meanwhile the sapphire surface exposed after the $MoS_2$ film is peeled off also shows an apparent water signal in the ATR-FTIR spectrum. These results verify the existence of a water layer at the $MoS_2$-sapphire interface. To show the possibility of the diffusion of water molecules into the interface, we performed density functional theory (DFT) calculations. The relevant details and results are given in Supplementary Section 10. The calculation results indicate that for the perfect $MoS_2$ films, water molecules could hardly diffuse into the film-substrate interface. However, the physical gap between $MoS_2$ and sapphire at some local edges or defects would be larger than the van der Waals gap, which have water molecules more easily infiltrate into the interface and further expand the physical gap. The expansion of the physical gap further supports a spontaneous infiltration of water molecules and formation of water nanolayer. The diffusion of water molecules and formation of water nanolayer undoubtedly decrease the interfacial adhesion strength. These phenomena and associated mechanisms have been reported in recent literatures[38,39].

To estimate the thickness of the interfacial water layer, the height profiles of $MoS_2$ thin films were measured by AFM at low and high RHs (Supplementary Fig. 11). Figure 3c shows that the thickness of a $MoS_2$ thin film increases by ~3 nm at 80 % RH compared to that in dry Ar (RH = 0), suggesting the existence of a 3-nm-thick water nanolayer at the film-substrate interface at high humidity. For $MoS_2$ thin films with pristine thicknesses ranging from 180 to 380 nm in dry air, the average thickness of this interfacial water nanolayer ranges from 2.9 to 3.9 nm at 80 % RH (inset of Fig. 3c), further proving that the water nanolayer is universal for our $MoS_2$-sapphire systems. In the case of such an interfacial water nanolayer, the capillary force is nonnegligible and contributes to the dynamic evolution of buckles. The capillary force, which is mainly ascribed to the negative pressure induced by the curvature of the water nanolayer (Fig. 3d, inset)[40], can be estimated by the following equation:

$$F_{cap} = \frac{\gamma_{H_2O} r_k (\cos\theta_{MoS_2} + \cos\theta_{Sapphire})^2}{d^2} \quad (1)$$

where $F_{cap}$ is the capillary force per unit area, $\gamma_{H_2O}$ is the surface tension of water, $r_k$ is the Kelvin radius of water, $\theta_{MoS_2}$ and $\theta_{sapphire}$ are the contact angles of water on $MoS_2$ and sapphire, respectively, and $d$ is the thickness of the interfacial water nanolayer. Consequently, the capillary force ranges from 46.4 to 11.6 MPa when the thickness of the water nanolayer varies from 2 to 4 nm at ~80% RH (Supplementary Section 11). Actually, the above equation overestimates the capillary force due to the neglect of dissolved gas. It has been reported that the critical negative pressure of water, which is the upper limit of the capillary force, is only several MPa because bubbles will start to nucleate beyond this pressure[41]. In our experiments, the circular blisters that appear at the propagated zones of arc buckles indicate the formation of bubbles in the water nanolayer (Supplementary Fig. 12), implying that the capillary force of the interfacial water nanolayer has reached its upper limit (several MPa). Considering that the capillary force that causes adhesion phenomena in microelectromechanical systems is typically tens of kPa[40,42], this MPa-level capillary force induced by the interfacial water nanolayer will dramatically alter the buckling morphology.

Based on the above results, we proposed a front delamination and tail re-adhesion mechanism to explain the formation of the SWLB (Fig. 3d). When the $MoS_2$ film is exposed to high humidity, water molecules diffuse through the film-substrate interface, resulting in delamination at the buckle tip when the humidity reaches a critical value (Fig. 3d). Simultaneously, an interfacial water nanolayer will form at the tail side of the buckle due to the higher effective humidity. The middle part of the arc buckle is a transition zone with saturated water vapor. This configuration is similar to the hydraulic fracture in rocks formed due to pressured liquids, where there also exists a tip cavity between the advancing fracture tip and the lagging fluid[43]. Moreover, the capillary force induced by the surface tension of the water will pull the suspended film re-adhere to the substrate (Fig. 3d, inset). Based on this front delamination and tail re-adhesion mechanism, the arc buckle will propagate forward integrally, forming the intriguing SWLB mode. It is noted that the re-adhesion process induced by the water nanolayer is essential for the formation of SWLB. Figure 3e shows a phase diagram to describe different buckling modes. When there is no re-adhesion process (i.e., $P_{re} = 0$), ordinary buckling modes will form and dominate the film deformation[16,21,44–47] (Fig. 3e). In this case, two parameters can be used to distinguish different delamination morphologies. The first parameter is the interface energy $G_c$, while the second one is the product of residual stress and film thickness $\sigma_r \cdot t$, representing the membrane force. In the phase diagram (Fig. 3e), the boundaries separating different buckling modes can be approximated by $G_c = c/(\sigma_r \cdot t)$, where $c$ is generally a constant[16]. For the boundary between TC and ring/circular regions (i.e., the blue solid line in Fig. 3e), c is fitted as 2500 $(J\,m^{-2})^2$. For the boundary between TC and web regions (i.e., the red solid line in Fig. 3e), c is obtained as 155 $(J\,m^{-2})^2$, which is close to that (156 $(J\,m^{-2})^2$) reported in a previous study[16]. Once the re-adhesion process is involved (i.e., $P_{re} = 1$) due to the introduction of a water nanolayer, SWLB will occur during film delamination, as shown in Fig. 3e.

## Simulations and theoretical modeling of the SWLB

The front delamination and tail re-adhesion mechanism suggests that the interfacial water nanolayer is crucial to the formation of the SWLB. We further used molecular dynamics (MD) simulation to analyse the effect of the water nanolayer on interfacial interactions. In our simulation, the simulated system consists of three layers of materials, a sapphire substrate, a layered $MoS_2$ film, and a 3-nm-thick water nanolayer between them (Fig. 4a). After equilibration, the $MoS_2$ film is separated from the interface, by which the corresponding traction-displacement curves can be simulated (see more details in Methods). Figure 4b, c plot the traction-displacement curves of the $MoS_2$/sapphire and $MoS_2$/$H_2O$/sapphire systems, respectively. It is clearly shown that the traction peaks at ~1.25 GPa when the separation displacement is ~0.4 nm for $MoS_2$/sapphire (Fig. 4b), which is a typical characteristic of interfaces with van der Waals interactions. When an ultrathin water layer is inserted between $MoS_2$ and sapphire, the traction is dramatically reduced, falling in the range of only 0–0.1 GPa (Fig. 4c). The interface adhesion energy can be derived from the traction-displacement curves, which are 0.249 $J\,m^{-2}$ and 0.093 $J\,m^{-2}$ for $MoS_2$/sapphire and $MoS_2$/$H_2O$/sapphire, respectively. The reduction in the interface adhesion energy suggests that the $MoS_2$ film is more prone to detaching from the substrate when there exists an interfacial water nanolayer, which is consistent with our experimental measurement (Fig. 3a). This result indicates that the $MoS_2$ film at the propagated zone is still weakly bonded onto the substrate due to the existence of the water nanolayer, which further supports the re-adhesion effect at the tail side.

Currently, there are some mechanical models that can accurately describe the common buckling modes of thin film, such as TC[18], web[23], circular[16], and ring-shaped buckling[21]. The predictions from these models are in good agreement with the corresponding experimental

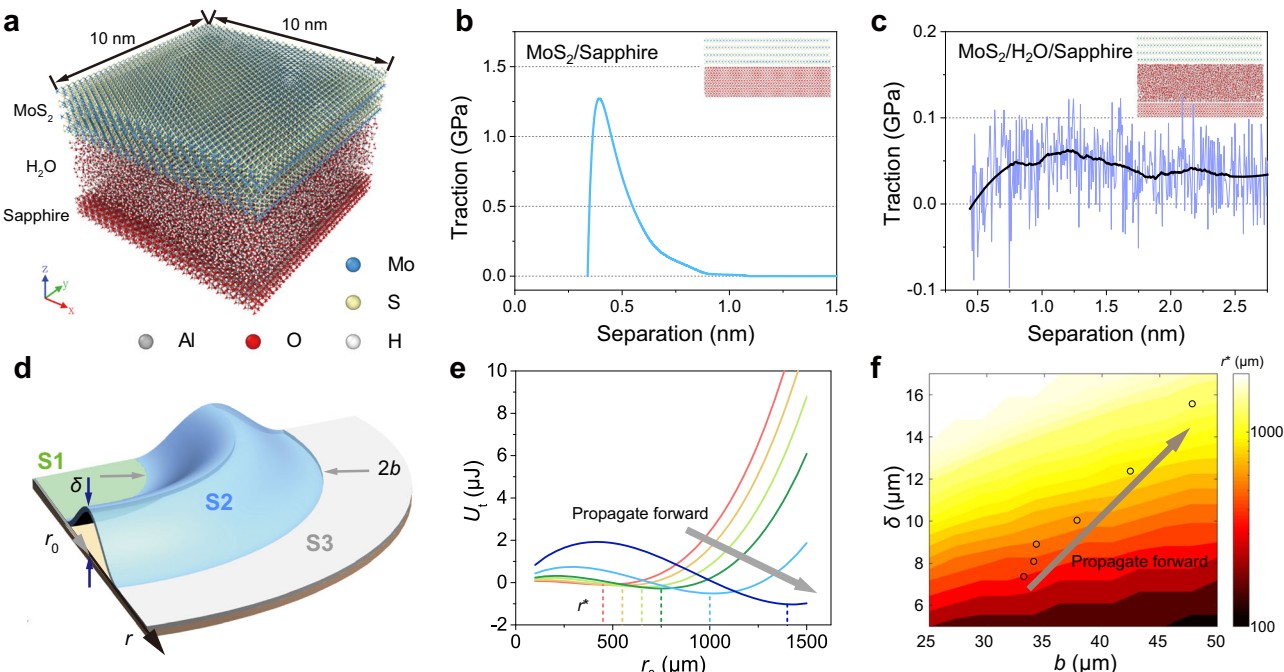

**Fig. 4 | MD simulations of interfacial interactions and theoretical modeling of SWLB propagation. a** Atomic model of MD simulations. **b**, **c** Traction-displacement curves obtained from MD for MoS$_2$/sapphire (**b**) and MoS$_2$/H$_2$O/sapphire (**c**). The data of MoS$_2$/H$_2$O/sapphire are smoothed using the Savitzky–Golay smoothing algorithm to show the trend (black line). **d** Schematic illustration of the theoretical model of SWLB propagation. $r_0$ represents the position of the arc buckle along the polar axis, and $2b$ and $\delta$ denote the width and height of the arc buckle in the theoretical model, respectively. S1, S2, and S3 are the inner flat area swept by the arc buckle, the buckling area where the film buckles and delaminates from the substrate, and the outer flat area where the film is pristine and the residual stress is not released, respectively. **e** Evolution of the total energy as a function of the radius of the SWLB with the six section profiles shown in Fig. 2g. **f** Equilibrium position contour of the SWLB with different sectional profiles predicted by our theoretical model. The six circles are the measured values of the radii in Fig. 2g. Source data are provided as a Source Data file.

results. However, these theoretical models cannot be applied to describe the current SWLB, because the SWLB propagation looks like solitary waves and is facilitated by water nanolayer between MoS$_2$ film and sapphire substrate, which is different from the common buckles in the thin film-substrate system. Furthermore, some theoretical models have been developed to describe the propagation of solitary waves in fluids[48,49] or solid structures[50,51], but these models cannot be used to describe the propagation of SWLB coupling with the delamination of thin film and readhesion facilitated by water nanolayer (i.e., FSI). To better understand the propagation behaviours of the SWLB mode, we developed a theoretical model by analysing the energy change during buckle steady-state propagation. In this model, the arc buckles are considered to be axisymmetric, that is, the buckling deformation along the circumferential direction is uniform. As shown in Fig. 4d, the film can be divided into three areas to calculate the corresponding energy: the inner flat area swept by the arc buckle (marked as S1), the buckling area where the film buckles and delaminates from the substrate (S2), and the outer flat area where the film is pristine and the residual stress is not released (S3). It is noted that the inner area (i.e. S1) is still bonded to the substrate due to the capillary effect of the interfacial water nanolayer, but the residual stress has been fully released. The energy of the S1 area is associated with the re-adhesion and formation of a new surface without residual stress. Therefore, the total energy $U_t$ of the MoS$_2$ film in the S1 and S2 areas can be expressed as:

$$U_t = S_2(U_b + U_s + G_{c2}) + S_1 \cdot G_{c1} - (S_1 + S_2)U_e \quad (2)$$

where $S_1$ and $S_2$ are the corresponding areas of S1 and S2, respectively; $U_b$ and $U_s$ are the bending energy and stretching energy in the S2 area, respectively; $G_{c2}$ is the energy release rate of the film-substrate interface in S2 area, $G_{c1}$ is the film-substrate re-adhesion energy

induced by the water nanolayer in S1 area, and $U_e$ is the elastic energy stored in these two areas due to the residual stress. In Eq. (2), the first term on its right side represents the increasing energies of the S2 area detached from the substrate, including the out-of-plane bending energy, the in-plane stretching energy, and the additional surface energy due to the formation of new surface in the S2 area. The second term means the energy associated with the readhesion and formation of a new surface in the S1 region. The third term reflects the releasing of elastic energies in both S1 and S2 areas due to the formation and propagation of arc buckle. We took $G_{c2} = 0.1 \, \text{J m}^{-2}$, $G_{c1} = 0.01 \, \text{J m}^{-2}$, $U_e = 0.223 \, \text{J m}^{-2}$, the elastic modulus $E_f = 29 \, \text{GPa}$, and the Poisson's ratio $v = 0.27$[23,52,53] (see more details in Methods and Supplementary Section 13).

As shown in Fig. 4e, the total energy $U_t$ as a function of coordinate $r_0$ can be derived at specific sectional profiles, which are obtained in Fig. 2g and described by the cosine function (Supplementary Fig. 13). The equilibrium states of the buckles correspond to the lowest energy points in a series of $U_t$-$r_0$ curves (Fig. 4e), and thus the equilibrium position $r^*$ of the SWLB can be theoretically determined for different sectional profiles. The relationship between the SWLB radius $r^*$ and its half-width $b$ is summarized in Supplementary Fig. 14. This shows that for a propagating SWLB, the half-width $b$ significantly increases with the expansion of $r^*$. Both the half-width $b$ and the height $\delta$ of the SWLB expand with the propagation of the SWLB, as shown in Fig. 4f. All of these trends are consistent with our experimental observations (Figs. 2h and 4f). Considering the dynamic effect of the propagation process, the buckles prefer to appear at some positions where the net energy change is negative, meaning that the possible positions of buckles fall into the purple zone shown in Supplementary Fig. 14. Two boundaries of such zones are determined by setting the net energy change to zero. This zone is close to the experimental results

(Supplementary Fig. 14), indicating that our theoretical model can capture the propagation and evolution of the SWLB. Notably, the rapid increase in the energy curve at large $r_0$ in Fig. 4e implies that the propagation of buckles with large radii is not energetically favorable and will stop spontaneously because large buckles require more bending energy. When the buckle propagation is stopped, the arc buckles with large radii usually suffer from fracture or breakage, including both failure of $MoS_2$ film (Supplementary Fig. 3a) and bifurcation of a large SWLB into small buckles (Supplementary Fig. 3b), due to the large tensile strain along the circumferential direction (Fig. 2i).

## Discussion

In summary, we discovered that the anomalous SWLB mode, the arc buckles, appears in a compressive strained $MoS_2$ film under high humidity. For this SWLB mode, the 3D profile of the buckles experiences an expansion process, which gradually releases the residual elastic strain energy in the film. Both in situ mechanical measurements and MD simulations suggest the weakening of interface interactions by humidity, and the interfacial water nanolayer, verified by in situ ATR-FTIR and AFM, provides a capillary effect for the readhesion of the film-substrate interface at the tail side of arc buckles. Based on these results, the front delamination and tail readhesion mechanism is proposed to explain the formation of the SWLB mode. Furthermore, the expansion morphologies and process of the SWLB are predicted by our theoretical model based on the energy change of buckle propagation. Our work discloses the dynamic propagation mechanism of the anomalous SWLB mode and demonstrates the significant influence of interfacial water nanolayers in solid thin films. This emerging SWLB mode will deepen the understanding of nanoscale FSI and shed light on both structural and functional applications of thin films.

Due to the significant difference in buckling height and propagating modes between SWLB and common buckles, this emerging buckling mode would have potential applications in sensors and energy conversion devices. First, compared with the tactile sensors developed with common buckles, the similar sensor enabled by the higher buckling height of the SWLB may have larger sensing range and higher sensitivity, which could be used to monitor vibrations, pressures, or humidity. Second, the height of the SWLB goes beyond the web buckles amplitude, indicating the releasing of more elastic strain energy. Therefore, the SWLB could be potentially used to fabricate energy conversion devices, in which controllable and stable propagation of the SWLB in nanoelectronic devices could enable self-powered function. Third, the large strain gradient built in the SWLB could induce flexo-photovoltaic or flexoelectric effects of $MoS_2$ films, providing a way to develop novel devices in deformed structures. However, before developing these proposed devices, it is reasonable to find a way to realize reversible deformation of the SWLB mode, which leaves an interesting topic for our future work.

## Methods
### Sample preparation
The Mo-polymer solution was prepared as follows: 2 g of ammonium molybdate tetrahydrate (Aladdin, ACS, 81–83% as $MoO_3$) was added to 20 mL of deionized water, and then 1 g of ethylenediaminetetraacetic acid (EDTA, Aladdin, 99.99%) and 2 g of ethylene imine polymer (PEI, Aladdin, M.W. 10000, 99%) was added to the mixed solution. After stirring for ~3 h, Amicon Ultra Centrifugal Filters with a 10000 molar weight-off membrane were used to purify the solution. The sapphire (0001) substrates were washed in acetone, isopropanol, and deionized water in an ultrasonic bath and then treated in piranha solution for 1 h. The Mo-polymer solution was spin-coated on substrates at 2500 rpm followed by sulfurization in a tube furnace at 850 °C for 30 min under an atmosphere of 10% $H_2$/90% Ar. Approximately 0.4 g of sulfur

powder (Aladdin, 99.99%) in a quartz boat was placed at the entrance of the quartz tube and maintained at ~190 °C by the heat belt.

### Humidity control and in situ observation
Homemade equipment was used to control the humidity quantitatively, including a container, moisture sensor, mass flow meter, conical flask, and Ar gas cylinder (Supplementary Fig. 2). A 99.999% Ar flow was used to bubble through the deionized water in the conical flask, and the flow rate was controlled by the mass flow meter to keep the container at the desired humidity. When the humidity was stable, the sample was rapidly placed in the container to encounter the moisture. This equipment was assembled with the probe station to observe the nucleation and propagation of the buckles. The surface topology of the buckles was measured by a laser-scanning confocal microscope (Olympus LEXT OLS 5000).

### Material characterizations
The thickness of the film was measured by an atomic force microscope (Bruker Multimode 8) under peak-force mode. Raman spectra were collected by a confocal Raman microscope (Horiba iHR550) with a 532-nm laser excitation, and a 50X long-working-distance lens was chosen to avoid contacting the sample. XPS measurements were conducted on a Thermo Fisher ESCALAB 250Xi with an excitation of Al $K_\alpha$ radiation. Another $SiO_2$/Si substrate was stuck to the surface of the $MoS_2$ film by double-sided adhesive tape, and then the $MoS_2$ film was peeled off from the sapphire for the XPS test. High-resolution transmission electron microscopy (HRTEM) and selected area electron diffraction (SAED) were obtained by a JEM-2100F at an acceleration voltage of 200 kV. The contact angle was measured by the sessile drop method on an optical contact angle measuring instrument (OCA25, Data-Physics), and drops of deionized water and diiodomethane were both 0.5 μL.

### Measurement of humidity-dependent properties of samples
The adhesion between the film and the substrate was measured by a tension sensor (INSTRON 5848 MicroTester), where the sample was fixed on the table by double-sided adhesive tape (3 M 9495LE-300LSE). The elastic modulus and hardness of the films were measured by a nanoindentation system (Keysight G200) using the continuous stiffness method, and the modulus of the films was extracted from the apparent modulus by the J. Hay model[54]. The absorbance spectra were acquired on a Nicolet iS50 FTIR Spectrometer (Thermo Scientific) with ATR accessory after the samples were blown with humidified air for ~15 min. In the measurement, the side of $MoS_2$ film was placed towards the ATR window. The $MoS_2$/PMMA sample used as the control sample in Fig. 3b was prepared by spin coating the PMMA solution (15%) on the $MoS_2$ film followed by peeling off the $MoS_2$ film from the sapphire substrate.

### Molecular dynamics simulations
MD simulations were performed via the large-scale atomic/molecular massively parallel simulator (LAMMPS) package[55]. The simulated system included a sapphire substrate, a layered $MoS_2$ film, and a 3-nm water layer between them, as shown in Fig. 4a. For comparison, a simulated system without a water nanolayer was constructed to calculate the interface energy between the $MoS_2$ film and sapphire substrate without a water nanolayer. Both systems have an in-plane dimension of $10 \times 10$ nm$^2$. The orientation between the $MoS_2$ film and sapphire substrate was set as the $x$ direction along [10$\bar{1}$0], $y$ direction along [$\bar{1}2\bar{1}0$], and $z$ direction along [0001]. In the simulations, the REBO potential was used to describe the interatomic interaction in the $MoS_2$ film[56], the CLAYFF potential for the sapphire substrate[57], and the TIP3P potential for water[58]. The entire system was equilibrated via energy minimization and subsequent free relaxation at 300 K. After equilibration, the density of the water nanolayer reached approximately

1.0 g cm$^{-3}$. Then, the $MoS_2$ film was pulled up at a constant speed of 10 nm s$^{-1}$ to simulate separation from the substrate, and the corresponding traction-displacement curves for systems with and without a water nanolayer were obtained by calculating the pulling stress and separation distance. The interfacial adhesion energies between the $MoS_2$ film and sapphire substrate with and without a water nanolayer were derived by integrating the areas covered by the corresponding traction-displacement curves.

## Theoretical Modeling

For simplicity, the arc buckles are considered to be axisymmetric, as shown in Fig. 4d. The film we studied consists of three areas, i.e., the inner propagated area (S1), the middle buckling area (S2), and the outer pristine area (S3). The buckling sectional profiles in Fig. 2g can be fitted well by the cosine functions (Supplementary Fig. 13). The total energy $U_t$ of the $MoS_2$ film in areas S1 and S2 are calculated based on Eq. (2). The bending energy $U_b$ and stretching energy $U_s$ in Eq. (2) are determined by solving the Föppl–von Kármán plate equation via the Airy stress function method (Supplementary Section 14). We took $G_{c2} = 0.1\,J\,m^{-2}$, $G_{c1} = 0.01\,J\,m^{-2}$, $U_e = 0.223\,J\,m^{-2}$, the elastic modulus $E_f = 29$ GPa, and the Poisson's ratio $v = 0.27$, and numerically solved Eq. (2). Finally, we plotted the $U_t$-$r_0$ curves for different sectional profiles (Fig. 4e). The lowest points of the $U_t$-$r_0$ curves reflect the equilibrium positions $r^*$ of the SWLB. Except for the measured arc buckles in Fig. 2g, other arc buckles with arbitrary sectional profiles (with various heights and widths) are also calculated to determine their equilibrium positions $r^*$ (the contour in Fig. 4f). More details are provided in Supplementary Section 14.

## Data availability

The data that supports the findings of this study are available in the manuscript and in the Supplementary Information section. Source data are provided with this paper as Source Data file. Source data are provided with this paper.

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

## Acknowledgements

This work was financially supported by the National Key R&D Program of China (2022YFA1203400) (K.L.), the Basic Science Center Project of NSFC under grant No. 52388201 (K.L.), and the National Natural Science Foundation of China under grant Nos. 51972193 (K.L.), 91963117 (X.L.), 11921002 (X.L.), and 11720101002 (X.L.). The authors acknowledge Olympus (Beijing) Inc. for help in the measurements of surface morphologies by laser scanning microscopy.

## Author contributions

E.W. and Z.X.X. contributed equally to this work. K.L. and X.L. designed the research. E.W. and H.R. prepared the samples. E.W. and J.G. set up the homemade humidity control equipment. E.W. and Z.Q.X. recorded the propagation of the buckles. E.W. and Y.S. measured the surface morphology. E.W., X.W. and C.L. characterized the samples. E.W., Z.X.X., B.W. and Z.Q.X. measured the humidity-dependent properties and discussed the mechanism. Z.X.X., Z.C., H.M. and X.L. performed theoretical analyses and molecular dynamics simulations. E.W., K.L., Z.X.X. and X.L. wrote and revised the manuscript. All authors discussed the results and contributed to the final version of the manuscript.

## Competing interests

The authors declare no competing interests.
