## [Peer Review File · Nature Communications]

Water nanolayer facilitated solitary-wave-like blisters in MoS₂ thin filmsREVIEWER COMMENTS

Reviewer #1 (Remarks to the Author):

Review comments on “Water nanolayer facilitated solitary-wave-like buckling propagation in MoS₂ thin films”.

The manuscript reports a discovery of solitary wave like blisters in MoS₂ films. The experimental procedure and observation techniques are clearly introduced. The formation mechanism of the SWLBs is also thoroughly explored. The interface toughness between the MoS₂ films and sapphire substrates is significantly compromised in the high humidity environment due to the formation a water layer of about 3nm thickness. The blisters nucleate at edges of the samples and propagate inwards to the central region of the samples. It is usually expected that telephone cord blisters (TCBs) are formed when secondary buckling occurs in the direction of blister axial direction at critical propagation length. However, the capillary force due to the high humidity environment ‘re-unites’ the separated surfaces together. The ‘reattachment’ has some distinct consequences. The secondary buckling is no longer possible resulting in no TCBs. The ‘reattachment’ is different from the intact surface in such a way that the residual strain energy in the film is released and transmitted to the wave front which requires a much larger amount of energy than that to drive a TCB. In addition, an arc wave front is observed as it needs much less energy than to drive a flat one. It nicely provides the self-strengthening mechanism for the formation of SWLB. However, when the ‘reattachment’ is unable to provide the amount energy required, SWLB will become stationary. In conclusion, the work is novel and significant, and the reviewer can strongly recommend it for publication and consider the following suggestions.

(1) It is advised to check that the blister height to see it is far beyond the post-buckling amplitude. The SWLB is driven by energy released from the ‘reattached’ flat films.

Consequently, it is advised that the title changes to ‘Water nanolayer facilitated solitary-wave-like blisters in MoS₂ thin films’.

(2) The developed mechanical model gives a good trend prediction for the stationary SWLB. Some latest existing mechanical model in open literature can be extended to give much more accurate prediction. The authors are advised to have a consideration, but not compulsory.

(3) The following suggestions are minor ones.

1) There are several occasions saying that the developed model gives perfect predictions. It is suggested to say reasonable predictions.

2) It is advised to use a consistent expression for interface fracture toughness $G-c$ instead of γ which is conventionally used for surface energy.

3) In Fig 4d in main article, replace w with δ to represent the amplitude of the blister.

4) Describe U_b and U_s in Eq. (2) in main article immediately after Eq. (2) and explain why Eq. (2) takes such expression.

5) At the bottom of page 5 of the main article, it says biaxial strain at interface. Please clarify if it is in the MoS₂ film or not.

6) The Poisson’s ratio of MoS₂ is missing in the work. The interface toughness value is given

as 0.1 J/m^2 . A reference source is recommended.

7) It is recommended that a clarification is given on R_t used in Fig. 2a in main article and r_0 in Fig. 4d also in main article.

8) It is recommended that a clarification is given for Fig. 2h in main article and Fig. 12 in supplementary.

Reviewer #2 (Remarks to the Author):

The authors report a very interesting discovery of solitary-wave-like buckling mode in a MoS₂ thin film/substrate system in a humid environment. They found that interfacial water nanolayers facilitate the delamination of films at the front side of the SWLB and a re-adhesion at the tail side. The expansion morphologies and process of the SWLB are predicted by an energy based theoretical model. The manuscript was well written and organized. I recommend that acceptance of this manuscript after addressing the following three issues.

1. The authors argued that the adhesion strength decreases due to the presence of nanolayer water at the interface between MoS₂ and sapphire substrate. However, for a hydrophilic film, its adhesion strength on a substrate without a pre-confined water layer will also decrease, and this usually results from the diffusion of water molecules during interfacial separation (Zhang et al. *Extreme Mechanics Letters*.2017, 16, 33-40). Besides, it is unclear in mechanism why water molecules would diffuse into the interface. Was there any defect or physical gap at the interface that was large enough to support a spontaneous infiltration of water molecules into and formation of nanolayer water, similar to that self-filling process of water molecules into a hydrophilic nanopore?

2. Water was found in the peeled MoS₂ by the ATR-FTIR spectrum and it was considered from nanolayer water at the interface. How to exclude the water was not from RH due to absorption after peeling? Were any residual water molecules from confined water layer observed at the peeled MoS₂ in simulations?

3. Buckling above a critical RH is led by the interfacial delamination and propagates with behavior of SWLB. Given the measured $\sim 0.35\%$ biaxial compressive strain that corresponded to a constant energy once released for buckling deformation, both the buckling mode and dimension of SWLB are supposed to remain. But it seems that the total interfacial area of buckled pattern increased in the results. Was it true? The authors mentioned that fracture or breakage occurred for buckles with large radii, what does that mean? Failure of MoS₂ or bifurcation of large buckles into small ones?

Reviewer #3 (Remarks to the Author):

The manuscript written by Enze Wang, et al. describes experimental demonstration of a new buckling mode in solid thin films and theoretical modeling of the formation mechanism. The authors focus on a unique ordered structure formed in nonlinear systems called solitary waves. Theoretical studies have shown that nonlinear effects in fluid-structure interactions can cause solitary waves, but there have been no studies that have demonstrated this experimentally. In this study, the authors fabricated MoS₂ films on sapphire substrates and humidified them precisely using an experimental system in which humidity can be elegantly controlled. Experimental results demonstrate the formation of solitary-wave-like-buckling (SWLB), which is different from typical meandering propagation modes such as telephone cord and web-like modes, due to the effect of interfacial water nanolayers. The interesting point of SWLB is that the deformed buckling propagates integrally, unlike the normal buckling mode in which the buckling region is pinned. To reveal these dynamics, the authors provide an excellent method to temporarily "freeze" the propagation of buckling by controlling the humidity in the experimental system and succeed in profiling the structure in detail. In addition, the adhesion between substrate and film, the humidity dependence of the mechanical properties of the film, and the wettability of the material are examined in detail, proposing a buckling mechanism facilitated by the capillary forces of the interfacial water nanolayers. The authors also reinforce the importance of interfacial water nanolayers in SWLB formation through molecular dynamics simulations, leading to the development of a theoretical model.

Overall, this work is interesting and solid, but a bit questionable in terms of novelty and impact for publication in Nature Communications. The experimental demonstration of the theoretically suggested new buckling mode is to be commended, but a more in-depth discussion or demonstration of how it might be applied would be of more interest to the readers. I therefore recommend the paper to be only publishable if its impact can be further validated.

Listed below are some additional comments:

Q1: The authors mention that the MoS₂ film undergoes biaxial compression due to thermal expansion mismatch with the substrate. It shows that this results in a blueshift of the Raman spectrum (Supplementary Fig. 1). Do the Raman spectra of the flat areas (such as S1 in Fig. 4d) where the SWLB propagated and energy was released match the Raman spectra of the MoS₂ film where strain was released?

Q2: Is it possible to present an application using this SWLB as reported by the authors (ACS Nano, 2022, 16, 9, 14157-14167)?

Reviewer #1 (Remarks to the Author):

Review comments on “Water nanolayer facilitated solitary-wave-like buckling propagation in MoS₂ thin films”.

[Comment] The manuscript reports a discovery of solitary wave like blisters in MoS₂ films. The experimental procedure and observation techniques are clearly introduced. The formation mechanism of the SWLBs is also thoroughly explored. The interface toughness between the MoS₂ films and sapphire substrates is significantly compromised in the high humidity environment due to the formation a water layer of about 3nm thickness. The blisters nucleate at edges of the samples and propagate inwards to the central region of the samples. It is usually expected that telephone cord blisters (TCBs) are formed when secondary buckling occurs in the direction of blister axial direction at critical propagation length. However, the capillary force due to the high humidity environment ‘re-unites’ the separated surfaces together. The ‘reattachment’ has some distinct consequences. The secondary buckling is no longer possible resulting in no TCBs. The ‘reattachment’ is different from the intact surface in such a way that the residual strain energy in the film is released and transmitted to the wave front which requires a much larger amount of energy than that to drive a TCB. In addition, an arc wave front is observed as it needs much less energy than to drive a flat one. It nicely provides the self-strengthening mechanism for the formation of SWLB. However, when the ‘reattachment’ is unable to provide the amount energy required, SWLB will become stationary. In conclusion, *the work is novel and significant, and the reviewer can strongly recommend it for publication and consider the following suggestions.*

[Response] We greatly appreciate your taking time to review our paper and evaluate our work. In our revised manuscript, we have performed more reliable experiments and calculations to support our statements. The following is a point-by-point response to the specific comments.

[Comment] (1) It is advised to check that the blister height to see it is far beyond the post-buckling amplitude. The SWLB is driven by energy released from the ‘reattached’ flat films. Consequently, it is advised that the title changes to ‘Water nanolayer facilitated solitary-wave-like blisters in MoS₂ thin films”.

[Response] We thank you very much for your valuable suggestions. To check whether the blister height of SWLB is beyond the post-buckling amplitude, we chose an arc buckle where web buckles form aside (Fig. R1a), and measured the heights of the arc and web buckles. As show in Fig. R1b, the height of the arc buckle (SWLB) was ~14 μm, much larger than the heights of the web buckles (post-buckling blisters) ranging from 2 to 3.5 μm. This reveals that the height of arc blister is far beyond the post-buckling amplitude, showing the release and accumulation of more residual elastic strain energy. In response to your comment, we added Fig. R1 as Supplementary Fig. 6, and above discussion on page 7 of the revised manuscript.

We also agreed with the reviewer and have changed the title into “Water nanolayer facilitated solitary-wave-like blisters in MoS₂ thin films”. We updated “solitary-wave-like buckles” as “solitary-wave-like blisters” and relevant descriptions in the revised manuscript.

Fig. R1 | Heights of different blisters. (a) Surface morphology of a SWLB and some adjacent web buckles characterized by a laser confocal microscope. Scale bar, 200 μm . (b) Height profile along the white line shown in (a).

[Comment] (2) The developed mechanical model gives a good trend prediction for the stationary SWLB. Some latest existing mechanical model in open literature can be extended to give much more accurate prediction. The authors are advised to have a consideration, but not compulsory.

[Response] We thank you very much for your helpful suggestion. Indeed, there have been some mechanical models (*Acta Mater.*, 2017, 125: 524-531; *ACS Nano*, 2019, 13: 3106-3116; *Acta Mater.*, 2017, 138: 1-9; *Thin Solid Films*, 2018, 651: 131-137) that can accurately describe the common buckling modes of thin films, including telephone-cord (TC), web, circular, and ring-shaped buckles. The predictions from these models are in good agreement with the corresponding experimental results. However, it is noted that the SWLB observed in MoS_2 thin films, which is facilitated by water nanolayer between MoS_2 film and sapphire substrate, is completely different from those buckles reported previously. Therefore, previous theoretical models for the common buckles cannot be applied to describe the current SWLB. Furthermore, there have been some theoretical models that can describe the propagation of solitary waves in fluids (*Phys. Lett. A*, 2021, 403, 127388; *J. Fluid Mech.*, 2022, 940, A15) or solid structures (*Phys. Rev. E*, 2022, 106, 054204; *Nat. Commun.*, 2019, 10, 5605). But these models cannot be used to describe the propagation of SWLB coupling the delamination of thin film and readhesion facilitated by water nanolayer. To better understand the propagation behaviors of SWLB mode in MoS_2 thin films, we developed a theoretical model by analyzing the energy evolution during the steady-state propagation of buckle. The results predicted from our theoretical model are close to the experimental results (Supplementary Fig. 13), indicating that our theoretical model can capture the propagation and evolution of the SWLB. Inspired by your suggestion and comment, we also expect to develop more accurate theoretical model to capture the SWLB feature in future study.

In response to your comment, we have added the following statements on page 13 and 14 in the revised manuscript to describe some existing theoretical models,

“Currently, there are some mechanical models that can accurately describe the common buckling modes of thin film, such as TC¹⁸, web²³, circular¹⁶, and ring-shaped buckling²¹. The predictions from these models are in good agreement with the corresponding experimental results. However, these theoretical models cannot be applied to describe the current SWLB, because the SWLB propagation looks like solitary waves and is facilitated by water nanolayer between MoS_2 film and sapphire substrate, which is different from the common buckles in the thin film-substrate system.

Furthermore, some theoretical models have been developed to describe the propagation of solitary waves in fluids^{48, 49} or solid structures^{50, 51}, but these models cannot be used to describe the propagation of SWLB coupling with the delamination of thin film and readhesion facilitated by water nanolayer (i.e., FSI).”

[Comment] (3) The following suggestions are minor ones.

1) There are several occasions saying that the developed model gives perfect predictions. It is suggested to say reasonable predictions.

[Response] We thank you very much for your suggestion. Following your suggestion, we changed the statement of “perfectly predicted” to “reasonably predicted” on page 4 in the revised manuscript.

[Comment] 2) It is advised to use a consistent expression for interface fracture toughness G_c instead of γ which is conventionally used for surface energy.

[Response] We thank you very much for your helpful suggestion. Following your suggestion, we have made the following modifications in the main text (pages 14, 15 and 19) and Supplementary Information (Section 13, page 16):

we replaced γ_w by G_{c1} to represent the film-substrate readhesion energy induced by the water nanolayer in S1 area, and also replaced γ with G_{c2} to represent the energy release rate of the film-substrate interface in S2 area.

[Comment] 3) In Fig 4d in main article, replace w with δ to represent the amplitude of the blister.

[Response] We thank you very much for your helpful suggestion. In light of your suggestion, we have replaced w with δ to represent the height of the SWLB in Fig. 4d. The revised figure is shown in Fig. R2.

Fig. R2 | Schematic illustration of the theoretical model of SWLB propagation. The width, height and propagation position of the SWLB are represented by $2b$, δ and r_0 , respectively.

[Comment] 4) Describe U_b and U_s in Eq. (2) in main article immediately after Eq. (2) and explain why Eq. (2) takes such expression.

[Response] We thank you very much for your helpful suggestions. In Eq. (2), U_b and U_s represent the bending energy and stretching energy in the buckling area, respectively. We have added the following statements on page 14 in the revised manuscript to describe U_b and U_s in Eq. (2),

“ U_b and U_s are the bending energy and stretching energy in the S2 area, respectively,”

The expression of Eq. (2) is used to describe the evolution of total energy of overall film-substrate system. By analyzing such energy evolution and seeking for the lowest energy point, we

can obtain the equilibrium state of SWLB and associated equilibrium position, and further describe its propagation (see Fig. 4e and Supplementary Fig. 13). Before buckling of MoS₂ film, both S1 and S2 areas (in Fig. R2a) store elastic strain energy due to the presence of residual stresses. When the MoS₂ film deforms into arc buckles propagating forward integrally, the elastic energy in both S1 and S2 areas are released, which is corresponding to the third term on the right side of Eq. (2). Due to the buckling of the S2 area, the increasing energies of the S2 area detached from the substrate include the out-of-plane bending energy, the in-plane stretching energy, and the additional surface energy due to formation of new surface in the S2 area. These energies are represented by the first term on the right side of Eq. (2). Similarly, the energy associated with the readhesion and formation of a new surface in the S1 region is represented by the second term on the right side of Eq. (2).

In the light of your suggestion, we have added the following statements on page 15 in the revised manuscript to explain why Eq. (2) takes such expression,

“In Eq. 2, the first term on its right side represents the increasing energies of the S2 area detached from the substrate, including the out-of-plane bending energy, the in-plane stretching energy, and the additional surface energy due to formation of new surface in the S2 area. The second term means the energy associated with the readhesion and formation of a new surface in the S1 region. The third term reflects the releasing of elastic energies in both S1 and S2 areas due to formation and propagation of arc buckle.”

[Comment] 5) At the bottom of page 5 of the main article, it says biaxial strain at interface. Please clarify if it is in the MoS₂ film or not.

[Response] We thank you very much for your comment. We confirmed that the biaxial strain is in the MoS₂ film. In response to your comment, we have added “a biaxial compressive strain of ~0.35% in the MoS₂ film” on page 5 in the revised manuscript.

[Comment] 6) The Poisson's ratio of MoS₂ is missing in the work. The interface toughness value is given as 0.1 J/m². A reference source is recommended.

[Response] We thank you very much for your helpful suggestion. The Poisson's ratio of MoS₂ is taken as 0.27 referring to the paper of Bertolazzi S. *et al* (*ACS Nano*, 2011, 5, 9703), and the interface toughness of 0.1 J/m² is referred to Sanchez D. A. *et al* (*Proc. Natl. Acad. Sci. USA*, 2018, 115, 7884), which is a typical value of TMDCs/sapphire system. In response to this comment, we added these two papers as references and indicated the Poisson's ratio on page 15 and 19 in the main text and Supplementary Section 13 in the revised manuscript.

[Comment] 7) It is recommended that a clarification is given on R_t used in Fig. 2a in main article and r_0 in Fig. 4d also in main article.

[Response] We thank you very much for your pointing out this issue. We would like to clarify that the symbol “ R_t ” used in Fig. 2a in the main article represents the transient curvature radius of SWLB during buckling propagation, which is experimentally measured. The symbol “ r_0 ” used in Fig. 4d represents the possible position of SWLB along the polar axis in the theoretical model. The equilibrium position “ r^* ” of SWLB is one of all possible position “ r_0 ” in the theoretical model, and means the position of SWLB with minimum energy. The measured position “ R_t ” in the experiments corresponds to the equilibrium position “ r^* ” in the theoretical model. Therefore, we compared the equilibrium position “ r^* ” predicted by the theoretical model with measured position “ R_t ” in the

experiments in Supplementary Fig. 14. To distinguish these two parameters, we expressed them in the different symbols.

In response to your comment, we have added the following statements in the corresponding positions to describe the meanings of R_t and r_0 , as well as other parameters,

In the caption of Fig. 2a,

“ R_t is the transient radius of the arc buckle at a specific time t , and R_f indicates the final radius of the arc buckle when the arc buckle no longer propagates. These two parameters can be measured experimentally.”

In the caption of Fig. 4d,

“ r_0 represents the position of the arc buckle along the polar axis, and $2b$ and δ denote the width and height of the arc buckle in the theoretical model, respectively. S1, S2, and S3 are the inner flat area swept by the arc buckle, the buckling area where the film buckles and delaminates from the substrate, and the outer flat area where the film is pristine and the residual stress is not released, respectively.”

[Comment] 8) It is recommended that a clarification is given for Fig. 2h in main article and Fig. 12 in supplementary.

[Response] We thank you very much for your pointing out this issue. Figure 2h shows the variations of the height δ_t and the half width b_t of the arc buckle with the transient curvature radius R_t . These data are obtained from the experimental measurements. Figure R3 (denoted as original Supplementary Fig. 12 in the previous version and Supplementary Fig. 14 in the revised version) shows a comparison between the equilibrium position (r^*) predicted by our theoretical model and the measured data (R_t) from our experiments for the arc buckles with different half widths (b_t). The red data points in Fig. 2h are the same as the black ones in Fig. R3. In response to your comment, we have changed the label “ r^* ” of vertical axis and the label “ b ” of horizontal axis into “ r^* or R_t ” and “ b_t ”, respectively. We have also updated the caption of Fig. R3, which is the Supplementary Fig. 14 in the revised Supplementary Information.

Fig. R3 | Comparison between the curvature radii (r^*) predicted by our theoretical model and the radii (R_t) from our experimental measurements for the arc buckles with different half widths (b_t).

Reviewer #2 (Remarks to the Author):

[Comment] The authors report a very interesting discovery of solitary-wave-like buckling mode in a MoS₂ thin film/substrate system in a humid environment. They found that interfacial water nanolayers facilitate the delamination of films at the front side of the SWLB and a re-adhesion at the tail side. The expansion morphologies and process of the SWLB are predicted by an energy based theoretical model. *The manuscript was well written and organized. I recommend that acceptance of this manuscript after addressing the following three issues.*

[Response] We greatly appreciate your taking time to review our paper and evaluate our work. In our revised manuscript, we have performed more reliable experiments and calculations to support our statements. The following is a point-by-point response to the specific comments.

[Comment] 1. The authors argued that the adhesion strength decreases due to the presence of nanolayer water at the interface between MoS₂ and sapphire substrate. However, for a hydrophilic film, its adhesion strength on a substrate without a pre-confined water layer will also decrease, and this usually results from the diffusion of water molecules during interfacial separation (Zhang et al. *Extreme Mechanics Letters*. 2017, 16, 33-40). Besides, it is unclear in mechanism why water molecules would diffuse into the interface. Was there any defect or physical gap at the interface that was large enough to support a spontaneous infiltration of water molecules into and formation of nanolayer water, similar to that self-filling process of water molecules into a hydrophilic nanopore?

[Response] We thank you very much for your insightful comments and providing this important literature. We agree with you that the adhesion strength between MoS₂ and sapphire substrate will decrease because of the diffusion of water molecules during interfacial separation, and that's exactly the way how our MoS₂ and sapphire substrate are separated. We carefully read the literature you suggested. In this literature (*Extreme Mechanics Letters* 2017, 16, 33), it was found that for a hydrophilic film water can reduce the peel-off force and promote the peeling process. Moreover, it has been experimentally reported that water molecules can diffuse into the interface through a physical gap between the hydrophilic SiO₂ substrate and graphene under high humidity conditions (*Nano Res.*, 2012, 5, 710), as you suggested. In our work, water molecules could also decrease the interfacial adhesion strength by diffusing into the MoS₂-substrate interface, where there should be a physical gap between the layered MoS₂ and the non-layered substrate. When the diffused water molecules accumulate there, a water nanolayer will form and further reduce the adhesion strength, triggering the SWLB.

To more clearly show the possibility of the diffusion of water molecules at the interface, we performed density functional theory (DFT) calculations via the Vienna Ab initio Simulation Package (VASP) program (*Comp. Mater. Sci.* 1996, 6, 15). The generalized gradient approximation with the Perdew–Burke–Ernzerhof type exchange-correlation functional (GGA-PBE) and projector augmented wave (PAW) method were adopted in all the calculations (*Phys. Lett. B* 1994, 50, 17953; *Phys. Rev. Lett.* 1996, 77, 3865). The plane-wave energy cutoff was fixed at 530 eV, and a $3 \times 3 \times 1$ Monkhorst–Pack grid was used for the k-point sampling. The van der Waals interactions were involved by using the DFT-D3 method of Grimme (*J. Phys. Chem. C* 2010, 132, 154104). A smearing of 0.1 eV was used. The convergence of energy and maximum force were set to 10^{-5} eV and 5×10^{-3} eV/Å, respectively. For the calculations of the adsorption energies of H₂O molecules inserted into the interface of MoS₂/sapphire, a (2×2) supercell of sapphire (001) with three atomic

layers was built. Then, a (3×3) unit cell of MoS₂ was adsorbed on the sapphire surface, with the contact distance d between MoS₂ and sapphire set to 3, 4, 5, 6, and 7 Å. An H₂O molecule was inserted between MoS₂ and sapphire.

The adsorption energy of the H₂O molecule is defined as

$$E_a = E_{\text{MoS}_2/\text{H}_2\text{O}/\text{sapphire}} - E_{\text{MoS}_2/\text{sapphire}} - E_{\text{H}_2\text{O}}$$

where $E_{\text{MoS}_2/\text{H}_2\text{O}/\text{sapphire}}$ and $E_{\text{MoS}_2/\text{sapphire}}$ are the energies of the MoS₂/sapphire model with and without the H₂O molecule, respectively. $E_{\text{H}_2\text{O}}$ is the energy of the H₂O molecule in vacuum.

As shown in Fig. R4, the adsorption energy is positive at the contact distance of 3 Å. As the van der Waals gap of CVD-grown MoS₂ on sapphire is usually about 3 Å (Xiang et al., *2D Mater.* 2020, 8, 025003), this calculation result shows that water molecules could hardly diffuse into the film-substrate interface for perfect MoS₂ films. However, the physical gap between MoS₂ and sapphire at some local edges or defects would be larger than the van der Waals gap. When the physical gap is larger than 3 nm, the adsorption energy of the H₂O molecule rapidly decreases to negative as the contact distance d increases up to 7 Å, which means that water molecules could more easily infiltrate into the interface and further expand the physical gap. The expansion of physical gap further supports a spontaneous infiltration of water molecules and formation of water nanolayer.

Fig. R4 | Calculated adsorption energies of a water molecule between MoS₂ and sapphire. (a) adsorption energies of a water molecule with different contact distances d ranging from 3 to 7 Å. (b) side view of MoS₂/H₂O/sapphire configuration.

In response to your comment, we have added the above DFT calculations into Supplementary Section 10 in Supplementary Information, added the following statements on page 9 and 10 in the revised manuscript to explain the mechanism of water molecule diffusing into the interface and the influence of nanolayer water on adhesion strength, and cited two literatures mentioned above (including the literature you suggested) as Refs. 38 and 39,

“To show the possibility of the diffusion of water molecules into the interface, we performed density functional theory (DFT) calculations. The relevant details and results are given in Supplementary Section 10. The calculation results indicate that for the perfect MoS₂ films, water molecules could hardly diffuse into the film-substrate interface. However, the physical gap between MoS₂ and sapphire at some local edges or defects would be larger than the van der Waals gap, which have water molecules more easily infiltrate into the interface and further expand the physical gap. The expansion of physical gap further supports a spontaneous infiltration of water molecules and formation of water nanolayer. The diffusion of water molecules and formation of water nanolayer undoubtedly decrease the interfacial adhesion strength. These phenomena and associated

mechanisms have been reported in recent literatures^{38, 39.}”

[Comment] 2. Water was found in the peeled MoS₂ by the ATR-FTIR spectrum and it was considered from nanolayer water at the interface. How to exclude the water was not from RH due to absorption after peeling? Were any residual water molecules from confined water layer observed at the peeled MoS₂ in simulations?

[Response] We thank you very much for your valuable comment. We guess that some ambiguous description in our manuscript may lead to a misunderstanding of the ATR-FTIR measurements. As shown in Fig. 3b, at a high humidity, both the MoS₂ film on sapphire and the exposed sapphire after removing the MoS₂ film have “liquid-like” water signals (top and middle spectra), while the MoS₂ film peeled off by PMMA does NOT show a water signal (bottom spectrum). The latter result has excluded the water adsorption on the surface of the MoS₂ film. These results verify that the water nanolayer exists at the MoS₂-sapphire interface, rather than on the surface of the MoS₂ film.

To avoid misunderstanding, we revised the discussion of Fig. 3b on page 9 of the revised manuscript as follows,

“Meanwhile the sapphire surface exposed after the MoS₂ film is peeled off also shows an apparent water signal in the ATR-FTIR spectrum.”

[Comment] 3. Buckling above a critical RH is led by the interfacial delamination and propagates with behavior of SWLB. Given the measured ~0.35% biaxial compressive strain that corresponded to a constant energy once released for buckling deformation, both the buckling mode and dimension of SWLB are supposed to remain. But it seems that the total interfacial area of buckled pattern increased in the results. Was it true? The authors mentioned that fracture or breakage occurred for buckles with large radii, what does that mean? Failure of MoS₂ or bifurcation of large buckles into small ones?

[Response] We thank you very much for your valuable comment. It is true that the total interfacial area of a single SWLB pattern increases during its propagation, which is quite different from other ordinary buckling modes like straight-sided and telephone-cord buckles. For straight-sided and telephone-cord buckles, their propagations are mainly driven by interfacial fracture at propagation tips, and the buckled structure cannot move, such that their buckling modes and dimension are mainly remained. However, the SWLB is driven by the front delamination and tail re-adhesion mechanism. During its propagation, the front side delaminates because water molecules reduce the adhesion strength, as mentioned in the response to the Reviewer’s comment 1, and the tail side reattaches to the substrate due to the capillary effect of water nanolayer, leading to a localized buckling propagation. This corresponds to a transfer of residual elastic strain energy stored in the MoS₂ film swept by the SWLB to the propagating SWLB, leading to the gradual increase in the elastic energy of the SWLB during propagation. As a result, the three-dimensional profile (R_i , δ_i , b_i) and the total interfacial area of SWLB increase with the increase of elastic energy.

For buckles with large radii (R_i), the fracture or breakage includes both failure of MoS₂ film (Supplementary Fig. 3a) and bifurcation of a large SWLB into small buckles (Supplementary Fig. 3b), due to the large tensile strain along the circumferential direction.

In response to your comment, we have revised the following discussion in the corresponding position,

On page 7 of the revised manuscript,

“With the increase in R_t , both the transient height δ_t and the half width b_t of the buckles increase (Fig. 2g), which means an increase in the total interfacial area and the elastic energy of the buckles.”

“The expansion of the three-dimensional profile (R_t , δ_t , b_t) and the total interfacial area of arc buckles suggests that the residual elastic strain energy stored in the MoS₂ film swept by the buckles may have transferred to the buckles”.

On page 16 of the revised manuscript,

“When the buckle propagation is stopped, the arc buckles with large radii usually suffer from fracture or breakage, including both failure of MoS₂ film (Supplementary Fig. 3a) and bifurcation of a large SWLB into small buckles (Supplementary Fig. 3b), due to the large tensile strain along the circumferential direction (Fig. 2i).”.

Reviewer #3 (Remarks to the Author):

[Comment] The manuscript written by Enze Wang, et al. describes experimental demonstration of a new buckling mode in solid thin films and theoretical modeling of the formation mechanism. The authors focus on a unique ordered structure formed in nonlinear systems called solitary waves. Theoretical studies have shown that nonlinear effects in fluid-structure interactions can cause solitary waves, but there have been no studies that have demonstrated this experimentally. In this study, the authors fabricated MoS₂ films on sapphire substrates and humidified them precisely using an experimental system in which humidity can be elegantly controlled. Experimental results demonstrate the formation of solitary-wave-like-buckling (SWLB), which is different from typical meandering propagation modes such as telephone cord and web-like modes, due to the effect of interfacial water nanolayers. The interesting point of SWLB is that the deformed buckling propagates integrally, unlike the normal buckling mode in which the buckling region is pinned. To reveal these dynamics, the authors provide an excellent method to temporarily "freeze" the propagation of buckling by controlling the humidity in the experimental system and succeed in profiling the structure in detail. In addition, the adhesion between substrate and film, the humidity dependence of the mechanical properties of the film, and the wettability of the material are examined in detail, proposing a buckling mechanism facilitated by the capillary forces of the interfacial water nanolayers. The authors also reinforce the importance of interfacial water nanolayers in SWLB formation through molecular dynamics simulations, leading to the development of a theoretical model.

Overall, *this work is interesting and solid, but a bit questionable in terms of novelty and impact for publication in Nature Communications*. The experimental demonstration of the theoretically suggested new buckling mode is to be commended, but a more in-depth discussion or demonstration of how it might be applied would be of more interest to the readers. I therefore recommend the paper to be only publishable if its impact can be further validated.

[Response] We greatly appreciate your taking time to review our paper and evaluate our work. In our revised manuscript, we have performed more reliable experiments to support our statements, and provided an in-depth discussion of potential applications of the SWLB. The following is a point-by-point response to the specific comments.

[Comment] Listed below are some additional comments:

Q1: The authors mention that the MoS₂ film undergoes biaxial compression due to thermal expansion mismatch with the substrate. It shows that this results in a blueshift of the Raman spectrum (Supplementary Fig. 1). Do the Raman spectra of the flat areas (such as S1 in Fig. 4d) where the SWLB propagated and energy was released match the Raman spectra of the MoS₂ film where strain was released?

[Response] We thank you very much for your valuable comment on Raman spectra. In the light of suggestion, we have collected the Raman spectra from an as-grown flat film area, the same area where the SWLB propagated, and the same area where strain was fully released after peeled off (Fig. R5). It is found that both E_{2g}¹ and A_{1g} Raman peaks of the flat area where the SWLB propagated well match those of the same area peeled off, which indicates that the strain in the MoS₂ film is completely released after the SWLB propagated.

Fig. R5 | Raman spectra of an as-grown flat MoS₂ film area (strained), the same area where the SWLB propagated (propagated), and the same area where strain was fully released after peeled off (released).

In response to this comment, we replaced Fig.S1f by Fig. R5 and added a description on page 5 and Supplementary Section 1 as follows,

“And the Raman spectrum collected from the film after SWLB propagated well match those of the same area peeled off, which indicates that the strain in the MoS₂ film is completely released after the SWLB propagated (Supplementary Fig. 1f)”.

[Comment] Q2: Is it possible to present an application using this SWLB as reported by the authors (ACS Nano, 2022, 16, 9, 14157-14167)?

[Response] We thank you very much for your insightful suggestion and comment. In current study, we mainly demonstrated the experimental discovery of the SWLB mode in a film-substrate system at a certain humidity condition and theoretically revealed the mechanism hidden behind. At this stage, the exploration of applications of the SWLB seems still premature because the SWLB deformation is localized and irreversible. In the light of your suggestion, we will discuss the potential applications of the SWLB.

The notable feature of the SWLB is that its three-dimensional profiles (R_t , δ_t , b_t) expands during propagation, which is quite different from other common buckles. The ultimate height of the SWLB is up to $\sim 14 \mu\text{m}$ (Fig. R1b), much higher than the height of other common buckles, including the web buckles ($\sim 2 \mu\text{m}$, Fig. R1b and ACS Nano, 2022, 16, 14157), TC buckles ($\sim 1.8 \mu\text{m}$, Acta Mater., 2017, 125, 524), circular buckles ($\sim 2.2 \mu\text{m}$, Acta Mater., 2017, 138, 1), and ring-shaped buckles ($\sim 1 \mu\text{m}$, Thin Solid Films, 2018, 651, 131). These results show that the SWLB may have continuously tunable size during its propagation and a wider-range strain gradient compared to other common buckles.

Due to the significant difference in buckling height and propagating modes between SWLB and common buckles, this new buckling mode would have potential applications in sensors and energy conversion devices. First, compared with the tactile sensors developed with common buckles (ACS Nano, 2022, 16, 14157), the similar sensor enabled by the higher buckling height of the SWLB may have larger sensing range and higher sensitivity, which could be used to monitor vibrations, pressures, or humidity. Second, the height of the SWLB goes beyond the web buckles amplitude,

indicating the releasing of more elastic strain energy. Therefore, the SWLB could be potentially used to fabricate energy conversion devices, in which controllable and stable propagation of the SWLB in nanoelectronic devices could enable self-powered function. Third, the large strain gradient built in the SWLB could induce flexo-photovoltaic or flexoelectric effects of MoS₂ films (*Nat. Nanotechnol.*, 2021, 16, 894-901), providing a way to develop novel devices in deformed structures. However, before developing these proposed devices, it is reasonable to find a way to realize reversible deformation of the SWLB, which is not the focus of this work and leaves an interesting topic for future work.

In response to your comment, we have added the following statements on page 17 in the revised manuscript to address the potential applications of the SWLB,

“Due to the significant difference in buckling height and propagating modes between SWLB and common buckles, this new buckling mode would have potential applications in sensors and energy conversion devices. First, compared with the tactile sensors developed with common buckles, the similar sensor enabled by the higher buckling height of the SWLB may have larger sensing range and higher sensitivity, which could be used to monitor vibrations, pressures, or humidity. Second, the height of the SWLB goes beyond the web buckles amplitude, indicating the releasing of more elastic strain energy. Therefore, the SWLB could be potentially used to fabricate energy conversion devices, in which controllable and stable propagation of the SWLB in nanoelectronic devices could enable self-powered function. Third, the large strain gradient built in the SWLB could induce flexo-photovoltaic or flexoelectric effects of MoS₂ films, providing a way to develop novel devices in deformed structures. However, before developing these proposed devices, it is reasonable to find a way to realize reversible deformation of the SWLB mode, which leaves an interesting topic for future work.”

REVIEWERS' COMMENTS

Reviewer #1 (Remarks to the Author):

The authors have made a great effort to address reviewer's comments successfully. It is a great pleasure to recommend it for a speedy publication.

Reviewer #2 (Remarks to the Author):

The authors addressed my questions and improved the manuscript nicely.

Reviewer #3 (Remarks to the Author):

Authors addressed the review comments politely. The manuscript has been much improved and is in a nice condition now. This paper is an important contribution and I recommend that it be accepted for publication in Nature Communications.

RESPONSE TO REVIEWERS' COMMENTS

[Comment] Reviewer #1 (Remarks to the Author):

The authors have made a great effort to address reviewer's comments successfully. It is a great pleasure to recommend it for a speedy publication.

[Response] We appreciate the reviewer's evaluation and all of the comments she/he raised during the review process.

[Comment] Reviewer #2 (Remarks to the Author):

The authors addressed my questions and improved the manuscript nicely.

[Response] We appreciate the reviewer's evaluation and all of the comments she/he raised during the review process.

[Comment] Reviewer #3 (Remarks to the Author):

Authors addressed the review comments politely. The manuscript has been much improved and is in a nice condition now. This paper is an important contribution and I recommend that it be accepted for publication in Nature Communications.

[Response] We appreciate the reviewer's evaluation and all of the comments she/he raised during the review process.